# Hyperglycaemia and Its Prognostic Value in Patients with COVID-19 Admitted to the Hospital in Lithuania

**DOI:** 10.3390/biomedicines12010055

**Published:** 2023-12-25

**Authors:** Lina Zabuliene, Ieva Kubiliute, Mykolas Urbonas, Ligita Jancoriene, Jurgita Urboniene, Ioannis Ilias

**Affiliations:** 1Institute of Clinical Medicine, Faculty of Medicine, Vilnius University, 03101 Vilnius, Lithuania; lina.zabuliene@mf.vu.lt; 2Clinic of Infectious Diseases and Dermatovenerology, Institute of Clinical Medicine, Faculty of Medicine, Vilnius University, 08661 Vilnius, Lithuania; ieva.kubiliute@santa.lt (I.K.); ligita.jancoriene@mf.vu.lt (L.J.); 3Faculty of Medicine, Vilnius University, 03101 Vilnius, Lithuania; 4Center of Infectious Diseases, Vilnius University Hospital Santaros Klinikos, 08661 Vilnius, Lithuania; jurgita.urboniene@santa.lt; 5Department of Endocrinology, Diabetes and Metabolism, Elena Venizelou Hospital, 11521 Athens, Greece

**Keywords:** coronavirus, SARS-CoV-2, COVID-19, glucose on admission, hyperglycaemia, in-hospital mortality

## Abstract

Background and objectives: Increased blood glucose levels atadmission are frequently observed in COVID-19 patients, even in those without pre-existing diabetes. Hyperglycaemia is associated with an increased incidence of severe COVID-19 infection. The aim of this study was to evaluate the association between hyperglycaemia at admission with the need for invasive mechanical ventilation (IMV) and in-hospital mortality in patients without diabetes who were hospitalized for COVID-19 infection. Materials and methods: This retrospective observational study was conducted at Vilnius University Hospital Santaros Clinics, Lithuania with adult patients who tested positive for severe acute respiratory syndrome coronavirus 2 SARS-CoV-2 and were hospitalized between March 2020 and May 2021. Depersonalized data were retrieved from electronic medical records. Based on blood glucose levels on the day of admission, patients without diabetes were divided into 4 groups: patients with hypoglycaemia (blood glucose below 4.0 mmol/L), patients with normoglycaemia (blood glucose between ≥4.0 mmol/L and <6.1 mmol/L), patients with mild hyperglycaemia (blood glucose between ≥6.1 mmol/L and <7.8 mmol/L), and patients with intermittent hyperglycaemia (blood glucose levels ≥7.8 mmol/L and <11.1 mmol/L). A multivariable binary logistic regression model was created to determine the association between hyperglycaemia and the need for IMV. Survival analysis was performed to assess the effect of hyperglycaemia on outcome within 30 days of hospitalization. Results: Among 1945 patients without diabetes at admission, 1078 (55.4%) had normal glucose levels, 651 (33.5%) had mild hyperglycaemia, 196 (10.1%) had intermittent hyperglycaemia, and 20 (1.0%) had hypoglycaemia. The oddsratio (OR) for IMV in patients with intermittent hyperglycaemia was 4.82 (95% CI 2.70–8.61, *p* < 0.001), and the OR was 2.00 (95% CI 1.21–3.31, *p* = 0.007) in those with mild hyperglycaemia compared to patients presenting normal glucose levels. The hazardratio (HR) for 30-day in-hospital mortality in patients with mild hyperglycaemia was 1.62 (95% CI 1.10–2.39, *p* = 0.015), while the HR was 3.04 (95% CI 2.01–4.60, *p* < 0.001) in patients with intermittent hyperglycaemia compared to those with normoglycaemia at admission. Conclusions: In COVID-19 patients without pre-existing diabetes, the presence of hyperglycaemia at admission is indicative of COVID-19-induced alterations in glucose metabolism and stress hyperglycaemia. Hyperglycaemia at admission in COVID-19 patients without diabetes is associated with an increased risk of invasive mechanical ventilation and in-hospital mortality. This finding highlights the importance for clinicians to carefully consider and select optimal support and treatment strategies for these patients. Further studies on the long-term consequences of hyperglycaemia in this specific population are warranted.

## 1. Introduction

Since the beginning of the severe acute respiratory syndrome coronavirus 2 (SARS-CoV-2) pandemic, diabetes mellitus has emerged as one of the most common comorbidities predisposing to coronavirus disease 2019 (COVID-19) and an important risk factor for hospitalization and mortality [1,2,3]. People with diabetes face a compromised immune system, making them more susceptible to severe outcomes of COVID-19. Diabetes is known for causing a pro-inflammatory syndrome, so patients with diabetes are more vulnerable to increased production of cytokines, which consequently lead to the deterioration of COVID-19 due to the development of acute respiratory distress syndrome and shock. Moreover, the hyperactivation of the coagulation cascade in COVID-19, particularly in the presence of pre-existing pro-thrombotic hypercoagulability associated with diabetes, intensifies the risk of severe thromboembolic complications [3]. Increased glucose levels at admission are frequently observed in COVID-19 patients, even in those without a history of diabetes.

Stress hyperglycaemia is a common condition in patients with any acute illness when the interaction of various cytokines and stress hormones leads to a state of insulin resistance and increased glucose production [4]. Inflammatory cytokines, such as interleukin-1 (IL-1), interleukin-6 (IL-6), and tumour necrosis factor-α (TNF-α), suppress insulin release and induce insulin resistance, and elevated levels of IL-6 promote hyperglycaemia by releasing glucose from hepatic glycogen reserves. Cortisol, catecholamines, glucagon, and growth hormone decrease insulin release by amplifying the activity of pancreatic alpha cells. Catecholamines further limit insulin binding, and, together with growth hormone, prevent insulin activation by suppressing tyrosine kinase activity. Moreover, catecholamines and glucocorticoids restrict glucose uptake in peripheral skeletal muscles through modulation of the glucose transporter type 4 [4]. 

Recent studies reported that stress hyperglycaemia increases the risk of mortality in critically ill patients [5] and patients with myocardial infarction [6] and stroke [7]. A single-centre study involving 1000 patients admitted to the intensive care unit found that in patients without diabetes, the risk of death increased by 20% for each 1 mmol/l increase in acute glycaemia [5]. Another study with 660 patients experiencing ST elevation myocardial infarction and treated with primary percutaneous coronary intervention showed significantly higher rates of mortality, cardiogenic shock, contrast-induced nephropathy, and the no-reflow phenomenon in the stress hyperglycaemia patient group [6]. A large study involving 8622 patients revealed that stress hyperglycaemia independently predicted severe neurological deficit within 1 year in patients with acute ischemic stroke, regardless of diabetes status. This association with mortality at 1 year is more pronounced in people without diagnosed or underlying diabetes [7]. In New Zealand, a study involving 739,152 ICU patients without a pre-existing diabetes diagnosis demonstrated a clear dose-response relationship between hyperglycaemia and hospital mortality, as well as an extended duration of hospital stay [8].

The appearance of hyperglycaemia in COVID-19 patients without diabetes likely indicates increased systemic stress as well. Furthermore, recent evidence from experimental studies suggests that SARS-CoV-2 infects human pancreatic β-cells and leads to morphological, transcriptional, and functional changes [9]. 

Hyperglycaemia is associated with a higher incidence of a severe course of COVID-19 infection, including the need for oxygen therapy [10,11], invasive mechanical ventilation (IMV) [10,11], and admission to the intensive care unit (ICU) [10,11,12] and has been found to be a predictor of in-hospital mortality in COVID-19 patients [12,13,14,15]. 

While extensive research has been conducted on the impact of hyperglycaemia in patients with diabetes and COVID-19, there remains a noticeable gap in our understanding of hyperglycaemia’s role in individuals without pre-existing diabetes. The aim of this study was to evaluate the association between hyperglycaemia at admission with the need for IMV and in-hospital mortality in patients without diabetes who were hospitalized for COVID-19 infection. By addressing this research gap, our study aims to contribute valuable insights that can inform clinical strategies and interventions tailored to patients without pre-existing diabetes, ultimately enhancing our ability to optimize care and improve outcomes in this specific population.

## 2. Materials and Methods

A single centre, retrospective observational cohort study was conducted at Vilnius University Hospital Santaros Klinikos (VUH SK), Vilnius, Lithuania [16].

### 2.1. Participants

Adult patients (18 years of age and older) who were hospitalized at VUH SK between March 2020 and May 2021 for confirmed COVID-19 infection and treated in the standard care unit, high dependency unit, or intensive care unit (ICU) were included in the study. The infection was verified based on a positive SARS-CoV-2 reverse transcriptase polymerase chain reaction result or a rapid antigen test conducted on a nasopharyngeal sample for symptomatic patients within 5 days from the onset of COVID-19 symptoms [16]. A total of 2561 hospitalized patients with COVID-19, who had their blood glucose tested during their hospital stay, were included in the analysis. Out of the 2561 patients hospitalized with COVID-19, 2054 had no pre-existing diabetes or a blood glucose concentration meeting the criteria for newly diagnosed diabetes during the hospital stay (Figure 1).

### 2.2. Data Collection and Variables

Depersonalized data were retrieved from the electronic medical records (EMR) of VUH SK and provided by the informatics and development centre in accordance with hospital-approved procedures [16].

Demographic variables included gender and age. Data on comorbidities included arterial hypertension (AH), coronary artery disease (CAD), congestive heart failure (CHF), diabetes mellitus, obesity, chronic obstructive pulmonary disease (COPD), chronic kidney disease (CKD), and previous stroke. Comorbidity data were extracted using the relevant codes from the International Statistical Classification of Diseases and Related Health Problems, Tenth Revision, Australian Modification (ICD-10-AM). AH was defined by the presence of comorbidities with ICD-10-AM codes I10, I11.0, and I11.9; CAD by the presence of comorbidities with ICD-10-AM codes I20.0, I20.2, I20.8, and I20.9; CHF by the presence of comorbidities with ICD-10-AM codes I50.0, I50.1, and I50.9; diabetes by the presence of comorbidities with ICD-10-AM codes E10 and E11; obesity by the presence of comorbidities with ICD-10-AM codes E66, E66.0, E66.2, E66.8, and E66.9; COPD by the presence of comorbidities with ICD-10-AM codes J44.0, J44.1, J44.8, and J44.9; CKD by the presence of comorbidities with ICD-10-AM codes N18.1, N18.2, N18.3, N18.4, N18.5, and N18.9; and previous stroke by the presence of comorbidities with ICD-10-AM code I69.If the condition was not documented in the patient’s EMR during hospitalization for COVID-19 infection, the patient was classified as not having that condition. Information regarding the utilization of antibiotics, systemic steroids, antivirals, IMV, length of hospital stay, and in-hospital mortality data was also extracted from depersonalized EMR.

Data from initial laboratory tests, including complete blood count, creatinine, urea, sodium, estimated glomerular filtration rate (eGFR), potassium, alanine aminotransferase (ALT), aspartate aminotransferase (AST), lactate dehydrogenase (LDH), C-reactive protein (CRP), ferritin, interleukin 6 (IL-6), D-dimer, and troponin I, were also retrieved and evaluated.The neutrophil-to-lymphocyte ratio (NLR) and AST to ALT ratio was calculated.

Data from blood glucose measurements conducted during hospitalization were obtained. Patients who had a blood glucose level of 11.1 mmol/L or higher during hospitalization were classified as having newly diagnosed diabetes mellitus. Based on the blood glucose level on the day of admission (if multiple blood glucose tests were performed within the first 24 h after admission, the mean glucose value was considered), patients without diabetes were divided into the following 4 groups: patients with hypoglycaemia were defined as those with blood glucose levels at admission below 4.0 mmol/L, patients with normoglycaemia had blood glucose levels at admission between ≥4.0 mmol/L and <6.1 mmol/L, patients with mild hyperglycaemia had blood glucose levels at admission was between ≥6.1 mmol/L and <7.8 mmol/L, and patients with intermittent hyperglycaemia had blood glucose levels atadmission between ≥7.8 mmol/L and <11.1 mmol/L.

Data from 2561 COVID-19 patients were analysed.

### 2.3. Main Outcome

The main outcome of the study was to evaluate the relationship between glycemia on admission in patients without diabetes and the severity of COVID-19 outcomes, including the need for mechanical ventilation and 30-day in-hospital mortality.

### 2.4. Statistical Analysis

Continuous and categorical variables were presented as medians (interquartile range [IQR]) and numbers (percentages, %), respectively. The Mann–Whitney U test was used to compare continuous variables, and the χ^2^ test was used to compare categorical variables. A multivariable binary logistic regression model was created to determine association of glucose levels and the need for IMV. The model included the need for IMV as the dependent variable, and age, gender, comorbidities which proportions statistically significantly differed between groups as predictors. We conducted survival analysis to assess the effects of glucose level on outcome within 30 days of hospitalization. We created a Cox proportional hazards regression model with in-hospital mortality as the dependent variable and age, gender, comorbidities with proportions that statistically significantly differed between groups, systemic steroids use, and treatment with remdesivir as predictors and plotted curves for survival stratified by glucose level. A two-sided *p*-value less than 0.05 was considered statistically significant. Analysis was performed using IBM Statistical Package for the Social Sciences software version 20.0 (IBM SPSS Statistics for Windows, Version 20.0, 2018, Armonk, NY, USA: IBM Corp.), and Microsoft Excel (Microsoft Corporation, 2018, Microsoft Excel, Available at: https://office.microsoft.com/excel, accessed on 18 December 2023) was used to produce figures.

## 3. Results

Among 2561 hospitalized adults, 55.1% were men. The median age was 60 years (IQR 49–70). Patients’ demographic, clinical characteristics, and initial laboratory parameters are presented in Table 1.

Reference values: Haemoglobin: 128–160 g/L (for males), 117–145 g/L (for females); WBC: 4.0–9.8 × 10^9^/L; neutrophils: 1.5–6.0 × 10^9^/L; lymphocytes: 1.0–4.0 × 10^9^/L; NLR—1–2; platelets: 140–450 × 10^9^/L; glucose: 4.2–6.1 mmol/L; creatinine: 64–104 µmol/L (for males), 49–90 µmol/L (for females); urea: 2.5–7.5 mmol/L; sodium: 134–145 mmol/L; potassium: 3.8–5.3 mmol/L; ALT: ≤40 U/L; AST: ≤40 U/L; AST to ALT ratio: <1; LDH: 125–243 U/L; CRP: <5 mg/L; ferritin: 25–350 µg/L (for men), 13–232 µg/L (for women); IL-6: 0–7 ng/L; D-dimer: <250 µg/L; and troponin I: <19 ng/L.

Out of a total of 2561 patients, 363 (14.2%) had a pre-existing diagnosis of diabetes before hospitalization, and 144 (5.6%) met the diagnostic criteria for newly diagnosed diabetes during their hospital stay. 

Glucose levels at admission were measured for 1945 patients without diabetes. Out of a total of 1945 patients without diabetes, 1078 (55.4%) had a normal glucose level at admission, 651 (33.5%) patients had mild hyperglycaemia at admission, and 196 (10.1%) patients had intermittent hyperglycaemia at admission. Additionally, 20 (1.0%) patients without diabetes had hypoglycaemia upon admission (Figure 1).

Patients with mild hyperglycaemia and intermittent hyperglycaemia were older compared to patients with normoglycaemia (Table 2). 

The prevalence of AH, CAD, CHF, obesity, CKD, and previous stroke was higher in patients with intermittent hyperglycaemia compared to patients with normoglycaemia. The prevalence of AH, CHF, and CKD was higher in patients with intermittent hyperglycaemia compared to patients with mild hyperglycaemia. 

The higher proportion of patients with intermittent hyperglycaemia required IMV compared to patients with mild hyperglycaemia (14.8% vs. 5.8%, *p* < 0.001) and normoglycaemia (14.8% vs. 2.7%, *p* < 0.001). 

A total of 1233 (64.1%) patients without diabetes received systemic steroids, with 1201 of them treated with dexamethasone for a course lasting 9 (IQR 6–10) days. A higher proportion of patients with mild hyperglycaemia (74.4% vs. 58.8%, *p* < 0.001) and intermittent hyperglycaemia (68.4% vs. 58.8%, *p* = 0.012) received systemic steroids compared to patients with normal glucose levels at admission.

A total of 632 (32.8%) patients without diabetes received remdesivir for a course lasting 5 (IQR 5–5) days. A higher proportion of patients with mild hyperglycaemia (37.2% vs. 30.2%, *p* = 0.003) received systemic steroids compared to patients with normal glucose levels at admission. However, there was no statistically significant difference between the mild and intermittent hyperglycaemia groups regarding the percentage of patients treated with remdesivir.

The highest in-hospital mortality rate was noted in patients with intermittent hyperglycaemia at admission reaching 22.4% compared to those with mild hyperglycaemia (9.4%) and normoglycaemia (5.3%) (Table 2).

Patients with intermittent hyperglycaemia on admission had higher levels of white blood cell (WBC) counts, neutrophil counts, NLR, platelet counts, creatinine, urea, ALT, AST, LDH, CRP, ferritin, IL-6, D-dimer, and troponin I compared to patients with normoglycaemia, whereas lymphocyte count, eGFR, potassium, and sodium concentration were significantly lower (Table 3). 

Reference values: haemoglobin: 128–160 g/L (for males), 117–145 g/L (for females); WBC: 4.0–9.8 × 10^9^/L; neutrophils: 1.5–6.0 × 10^9^/L; lymphocytes: 1.0–4.0 × 10^9^/L; NLR—1–2; platelets: 140–450 × 10^9^/L; glucose: 4.2–6.1 mmol/L; creatinine: 64–104 µmol/L (for males), 49–90 µmol/L (for females); urea: 2.5–7.5 mmol/L; sodium: 134–145 mmol/L; potassium: 3.8–5.3 mmol/L; ALT: ≤40 U/L; AST: ≤40 U/L; AST to ALT ratio: <1; LDH: 125–243 ULl; CRP: <5 mg/L; ferritin: 25–350 µg/L (for men), 13–232 µg/L (for women); IL-6: 0–7 ng/L; D-dimer: <250 µg/L; and troponin I: <19 ng/L.

Patients with mild hyperglycaemia had higher levels of WBCs, neutrophil counts, NLR, creatinine, urea, ALT, AST, LDH, CRP, ferritin, IL-6, D-dimer, and troponin I compared to patients with normoglycaemia, whereas lymphocyte counts and eGFR, potassium, and sodium concentrations were significantly lower (Table 3).

Among patients with intermittent hyperglycaemia on admission, the odds ratio (OR) for IMV was 4.82 (95% CI 2.70–8.61, *p* < 0.001), while the OR was 2.00 (95% CI 1.21–3.31, *p* = 0.007) in those with mild hyperglycaemia compared to patients presenting normal glucose levels at admission. Other risk factors associated with the need for IMV were male gender, obesity, and CHF (Figure 2).

Cox regression analysis revealed that the hazard ratio (HR) for 30-day in-hospital mortality in patients with mild hyperglycaemia on admission was 1.62 (95% CI 1.10–2.39, *p* = 0.015), whilethe HR was 3.04 (95% CI 2.01–4.60, *p* < 0.001) in patients with intermittent hyperglycaemia compared to those with normoglycaemia (Figure 3). 

Other risk factors associated with increased HR for 30 days in-hospital mortality were age, obesity, CHF, and previous stroke. Treatment with Remdesivir was associated with reduced HR (0.56 (95% CI 0.37–0.85, *p* = 0.006) for 30-day in-hospital mortality) (Table 4).

## 4. Discussion

Infection induces significant alterations in glucose metabolism. During infection, there is an upsurge in glucose production attributed to hepatic gluconeogenesis and glycogenolysis accompanied by reduced peripheral glucose uptake due to decreased blood flow in muscles (reversible insulin resistance) and increased anaerobic glycolysis due to hypoxia, resulting in elevated blood glucose levels, regardless of pre-existing diabetes status. Such a response is believed to be influenced by a combination of neurohumoral changes, activation of counterregulatory hormones (glucagon, epinephrine, cortisol, and growth hormone), excessive release of proinflammatory cytokines (IL-6, IL-1β, and tumour necrosis factor (TNF)-α), reactive oxygen species production, impaired immune cell function, and the release of lipid mediators [17,18,19,20].

Various viral infections, including cytomegalovirus, enteroviruses, Epstein-Barr virus, hepatitis B and C virus, human immunodeficiency virus, and influenza, use distinct mechanisms to promote hyperglycaemia and can directly affect glucose metabolism and insulin signaling pathways, leading to dysregulation of glucose homeostasis [21]. Recent findings have shed light on the interplay among viral factors, immune response, inflammation, pancreatic function, and the potential role of insulin resistance in driving the occurrence of hyperglycaemia during COVID-19 infection [21,22,23,24,25].

Our study demonstrated that 43.6% of patients without pre-existing diabetes hospitalized for COVID-19 infection had increased blood glucose levels atadmission. Data on the prevalence of hyperglycaemia in COVID-19 patients without diabetes varies and depends on the definition of hyperglycaemia and the population studied. Our study data are comparable to data from Montefusco et al., who reported that 46% of COVID-19 patients had hyperglycaemia, defined as blood glucose levels between 5.6 mmol/L and 11.1 mmol/L or two blood glucose measurements of >5.6 mmol/Land < 7.0 mmol/L [26]. Mamtani et al. reported that 20.6% of COVID-19 patients without pre-existing diabetes had blood glucose levels at admission ≥ 7.78 mmol/L [8]. Zhang et al. found that 20% of COVID-19 patients were in a state of impaired fasting glucose (defined as fasting blood glucose between ≥5.6 mmol/L and <7.0 mmol/L) [15]. 

The prognostic importance of hyperglycaemia on the outcomes of COVID-19 infection has been emphasized in multiple studies. Hyperglycaemia atadmission in patients without diabetes is associated with a higher incidence of a severe course of COVID-19 infection, need for IMV, admission to the ICU [10,11,12,24,25,26,27,28,29,30], and higher mortality [13,14,27,29,31,32]. Our study showed that a higher proportion of patients with intermittent hyperglycaemia at admission (14.8%) required IMV compared to patients with mild hyperglycaemia (5.8%) and patients with normoglycaemia (2.7%). The risk of IMV in patients with intermittent hyperglycaemia was 4.82-fold higher than in those with normoglycaemia. Iacobellis et al. noted that average blood glucose level at admission was the strongest independent predictor of bilateral pulmonary opacities suggestive of acute respiratory distress in the radiographic imaging of COVID-19 patients [27]. Fadine et al. reported that for every 2 mmol/L increase in glucose level at admission, the probability of severe COVID-19 increased by approximately15% independently from any other clinical-biochemical variables [28]. We determined that the HR for 30-day in-hospital mortality in patients with mild hyperglycaemia was 1.62, while the HR was 3.04in patients with intermittent hyperglycaemia compared to those with normoglycaemia. A meta-analysis including sixteen observational studies with 6386 COVID-19 patients demonstrated that the group of patients with hyperglycaemia at admission had an increased risk of mortality compared to the control group with euglycaemia (OR = 3.45, 95% CI 2.26–5.26) [30]. Morse et al. retrospectively showed that among patients without a pre-existing diabetes diagnosis, any hyperglycaemic value was associated with a substantial increase in the odds of mortality (OR = 3.07, 95% CI 2.79–3.37) compared to patients without hyperglycaemia [33]. 

The association between hyperglycaemia and increased mortality in COVID-19 patients is likely multifactorial, involving inflammation, immune dysfunction, vascular damage, and underlying comorbidities. Hyperglycaemia is known to contribute to a state of systemic inflammation, which can worsen the severity of infection and outcome in COVID-19 patients [34]. High blood glucose levels have been shown to impair the function of various immune cells, such as T cells and macrophages, thereby compromising the immune response against the virus [35]. Hyperglycaemia can contribute to endothelial dysfunction and arteriopathy, leading to complications such as thrombosis and impaired oxygen delivery, which can further exacerbate the severity of COVID-19 and increase the mortality risk [36]. Furthermore, individuals with newly developed hyperglycaemia during COVID-19 infection may be older and have underlying comorbidities, which can increase vulnerability and in-hospital mortality [34,37,38,39]. In our study, patients with mild or intermittent hyperglycaemia were older and had more comorbid conditions compared to patients with normoglycaemia. Furthermore, we observed elevated troponin I levels at admission in patients with intermittent hyperglycaemia.

The cytokine storm is a characteristic feature of COVID-19 infection and is strongly associated with disease severity and mortality [40]. The laboratory features of cytokine storms include haematological anomalies, such as leucocytosis or leukopenia, thrombocytopenia, and disseminated intravascular coagulation; high fibrinogen levels; elevated IL-6; and general markers of end-organ dysfunction [41]. We observed a significantly lower lymphocyte count, higher NLR, and higher IL-6 and CRP levels in patients with mild and intermittent hyperglycaemia, presuming that the inflammatory response is more pronounced in patients with increased glucose levels at admission. Our results are comparable to the findings of Coppelli et al., as we also observed a higher neutrophil count, lower lymphocyte count, and higher CRP in patients without known diabetes with hyperglycaemia (glucose level ≥ 7.78 mmol/L) at admission compared to patients with normoglycaemia [29].

Beyond the immediate impact on glucose metabolism, investigating the long-term consequences of COVID-19 infection and potential implications for metabolic health emerges as a crucial direction of exploration. Montefusco et al. highlighted the profound impact of COVID-19 on disrupting insulin signalling and beta cell function. They reported significant alterations in hormone profiles, both at basal levels and after stimulation testing, demonstrating elevated insulin, proinsulin, and C-peptide levels in patients who had recovered from COVID-19 compared to healthy controls [26]. In a recent meta-analysis that scrutinized the occurrence of new-onset diabetes and hyperglycaemia after COVID-19 infection in patients without pre-existing diabetes, the proportions of patients with new-onset diabetes was 3% and new-onset hyperglycaemia was 30%. Additionally, the meta-analysis found that new-onset diabetes and hyperglycaemia were 1.75 times higher in COVID-19 patients compared to non-COVID-19 patients [42]. Recommendations have been made for COVID-19 hyperglycaemic patients, emphasizing the need for follow-up at the first month and at intervals of 3–6 months during the first year post-discharge to discern whether hyperglycaemia is permanent or transient [43].

This study has several limitations. Comorbidities were identified based on ICD-10-AM coding, which may have resulted in some incomplete attributions. Additionally, information about fasting status on admission and glycated haemoglobin (HbA1c) was not available for all patients, potentially biasing the detection of pre-existing hyperglycaemia. Furthermore, the absence of data on body mass index limited our ability to accurately determine obesity in patients. Lastly, we were unable to control for the pre-admission use of glucocorticoids and other medications, which could have influenced our findings. An additional challenge relates to missing information on the duration of COVID-19 symptoms before hospital admission. While our research was conducted at the largest hospital in the capital of Lithuania, limiting the study to a single centre introduces the potential for selection bias and reduces the generalizability of our findings. The healthcare practices, patient demographics, and treatment protocols at this specific centre may not fully represent the broader population. An additional limitation of our retrospective study is the lack of information on the long-term consequences of hyperglycaemia in COVID-19 patients. As a retrospective analysis primarily focused on immediate outcomes during hospitalization, we did not have the opportunity to explore the extended effects of hyperglycaemia beyond the acute phase of the disease. Nonetheless, our study has several strengths, including the comprehensive clinical characterization of all patients who were hospitalized for COVID-19 infection over a period of more than one year in the largest university hospital in the country, as well as the evaluation of multiple laboratory parameters. 

## 5. Conclusions

In COVID-19 patients without pre-existing diabetes, the presence of hyperglycaemia at admission is indicative of COVID-19-induced alterations in glucose metabolism and stress hyperglycaemia. Hyperglycaemia at admission in COVID-19 patients without diabetes is associated with an increased risk of invasive mechanical ventilation and in-hospital mortality. This finding highlights the importance for clinicians to carefully consider and select optimal support and treatment strategies for these patients. Further studies on the long-term consequences of hyperglycaemia in this specific population are warranted.

## Figures and Tables

**Figure 1 biomedicines-12-00055-f001:**
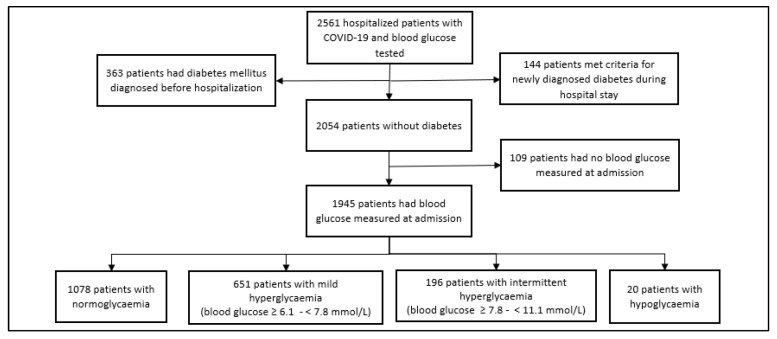
Distribution of patients hospitalized with COVID-19 infection.

**Figure 2 biomedicines-12-00055-f002:**
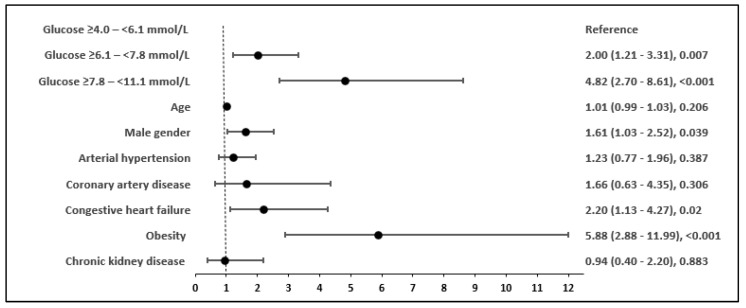
Risk factors and odds ratios (OR) associated with the need for invasive mechanical ventilation in COVID-19 patients without diabetes.

**Figure 3 biomedicines-12-00055-f003:**
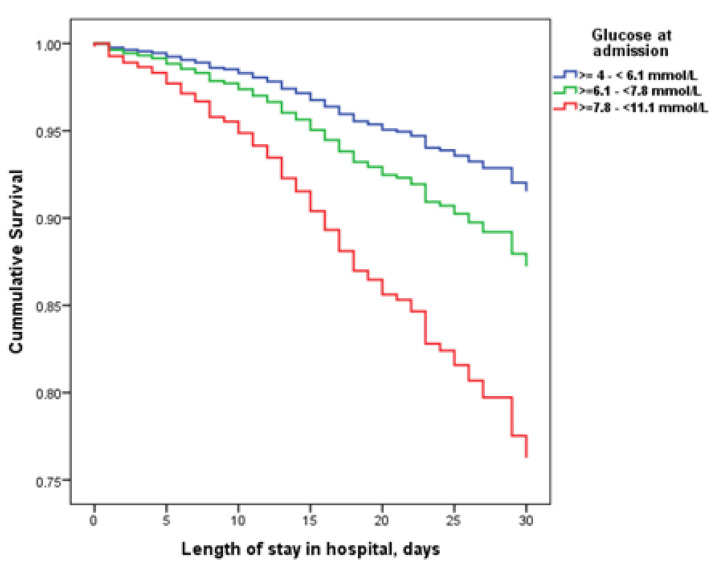
Survival of hospitalized patients without diabetes stratified by glycaemia levels atadmission.

**Table 1 biomedicines-12-00055-t001:** Demographic, clinical, and initial laboratory characteristics of patients hospitalized with COVID-19 infection.

Demographic and Clinical Characteristic (2561 Patients)	N (%)	Laboratory Characteristics	N	Median (IQR)
Age, years, median (IQR)	60 (49–70)	Haemoglobin, g/L	2561	138 (124–149)
Male	1412 (55.1)	WBC, ×10^9^/L	2561	6.48 (4.84–9.03)
Female	1149 (44.9)	Neutrophils, ×10^9^/L	2561	4.76 (3.30–7.13)
Any concommitant condition	1252 (48.9)	Lymphocytes, ×10^9^/L	2561	1 (0.70–1.40)
Arterial hypertension	983 (38.4)	NLR	2558	4.70 (2.86–8.05)
Coronary artery disease	90 (3.5)	Platelets, ×10^9^/L	2561	198 (153–258)
Congestive heart failure	198 (7.7)	Glucose, mmol/L	2561	6.12 (5.43–7.28)
Diabetes mellitus	363 (14.2)	Creatinine, µmol/L	2557	82 (67–105)
Obesity	123 (4.8)	Urea, mmol/L	2371	5.73 (4.12–8.74)
COPD	42 (1.6)	eGFR, mL/min/1.73 m^2^	2513	83 (58.95–96)
Chronic kidney disease	205 (8.0)	Sodium, mmol/L	2551	140 (137–143)
Previous stroke	32 (1.2)	Potassium, mmol/L	2551	4.20 (3.90–4.60)
Invasive mechanical ventilation	192 (7.5)	ALT, U/L	2500	31.42 (19.65–52)
Antibiotics use	1863 (72.7)	AST, U/L	2485	36 (26–57)
Antivirals (remdesivir)	808 (31.6)	AST to ALT ratio	2483	1.17 (0.87–1.67)
Systemic steroids	1630 (63.6)	LDH, U/L	2302	303 (236–408.04)
In-hospital mortality	313 (12.2)	CRP, mg/L	2558	62.15 (22.78–124.88)
Length of hospital stay, days, median (IQR)	11 (7–16)	Ferritin, µg/L	2367	479.85 (236–1009.44)
		IL-6, ng/L	2254	29.60 (14.40–57.30)
		D-dimer, µg/L	2344	510 (305–985)
		Troponin I, ng/L	2122	10 (5–26)

ALT—alanine aminotransferase; AST—aspartate aminotransferase; COPD—chronic obstructive pulmonary disease, CRP—C-reactive protein; eGFR—estimated glomerular filtration rate; IL-6—interleukin 6; IQR—interquartile range; LDH—lactate dehydrogenase; N—number; NLR—neutrophil-to-lymphocyte ratio; WBC—white blood cell count.

**Table 2 biomedicines-12-00055-t002:** Demographic and clinical characteristics of COVID-19 patients without diabetes based on glycaemia levels on admission.

Demographic and Clinical Characteristic	Normoglycaemia N = 1078	Mild Hyperglycaemia, N = 651	Intermittent Hyperglycaemia, N = 196	*p*-Value ^1^	*p*-Value ^2^	*p*-Value ^3^
Age in years, median (IQR)	55 (43–66)	60 (50–68)	67 (59–76)	<0.001	<0.001	<0.001
Male	604 (56.0%)	379 (58.2%)	108 (55.1%)	0.373	0.810	0.439
Female	474 (44.0%)	272 (41.8%)	88 (44.9%)	0.373	0.810	0.439
Any underlying condition	373 (34.6%)	307 (47.2%)	119 (60.7%)	<0.001	<0.001	0.001
Arterial hypertension	299 (27.7%)	246 (37.8%)	96 (49.0%)	<0.001	<0.001	0.005
Coronary artery disease	23 (2.1%)	19 (2.9%)	10 (5.1%)	0.304	0.016	0.141
Congestive heart failure	48 (4.5%)	38 (5.8%)	21 (10.7%)	0.200	<0.001	0.019
Obesity	24 (2.2%)	22 (3.4%)	12 (6.1%)	0.149	0.002	0.086
COPD	12 (1.1%)	10 (1.5%)	2 (1.0%)	0.447	1.000	0.743
Chronic kidney disease	58 (5.4%)	30 (4.6%)	18 (9.2%)	0.479	0.039	0.015
Previous stroke	8 (0.7%)	5 (0.8%)	5 (2.6%)	1.000	0.037	0.057
Invasive mechanical ventilation	29 (2.7%)	38 (5.8%)	29 (14.8%)	0.001	<0.001	<0.001
Antibiotics	763 (70.8%)	486 (74.7%)	141 (71.9%)	0.081	0.742	0.447
Antivirals (remdesivir)	326 (30.2%)	242 (37.2%)	64 (32.7%)	0.003	0.500	0.248
Systemic steroids	634 (58.8%)	465 (71.4%)	134 (68.4%)	<0.001	0.012	0.409
In-hospital mortality	57 (5.3%)	61 (9.4%)	44 (22.4%)	0.001	<0.001	<0.001
Length of hospital stay, days, median (IQR)	10 (7–14)	11 (8–16)	11.50 (7–16)	0.001	0.059	0.996

1—*p* value comparing patients with normoglycaemia vs. patients with mild hyperglycaemia. 2—*p* value comparing patients with normoglycaemia vs. patients with intermittent hyperglycaemia. 3—*p* value comparing patients with mild hyperglycaemia vs. patients with intermittent hyperglycaemia.

**Table 3 biomedicines-12-00055-t003:** Initial laboratory characteristics of COVID-19 patients without diabetes based on glycaemia levels upon admission.

Laboratory Characteristics	Normoglycaemia		Mild Hyperglycaemia		Intermittent Hyperglycaemia		*p*-Value ^1^	*p*-Value ^2^	*p*-Value ^3^
	n	Value, Median (IQR)	n	Value, Median (IQR)	n	Value, Median (IQR)			
Haemoglobin, g/L	1078	139 (125–150)	651	140 (127–151)	196	141 (124–151)	0.120	0.693	0.621
WBC, ×10^9^/L	1078	5.87 (4.43–7.62)	651	6.54 (5.05–8.91)	196	8.22 (6.07–11.68)	<0.001	<0.001	<0.001
Neutrophils, ×10^9^/L	1078	4.10 (2.90–5.80)	651	5 (3.60–7.16)	196	6.60 (4.50–9.90)	<0.001	<0.001	<0.001
Lymphocytes, ×10^9^/L	1078	1.07 (0.80–1.46)	651	0.93 (0.66–1.26)	196	0.90 (0.60–1.33)	<0.001	<0.001	0.732
NLR	1076	3.73 (2.45–5.91)	650	5.20 (3.30–8.46)	196	7.47 (4.50–12.54)	<0.001	<0.001	<0.001
Platelets, ×10^9^/L	1078	194 (151–254)	651	195 (153–253)	196	213.50 (168.25–277.75)	0.832	0.001	0.003
Glucose at admission, mmol/L	1078	5.46 (5.09–5.77)	651	6.60 (6.33–7.06)	196	8.70 (8.16–9.57)	<0.001	<0.001	<0.001
Creatinine, µmol/L	1075	78.2 (65–96)	651	81.01 (68–98.93)	196	87 (70.25–130.93)	0.015	<0.001	0.001
Urea, mmol/L	970	4.98 (3.71–6.68)	608	5.60 (4.10–7.82)	182	7.13 (4.88–14.15)	<0.001	<0.001	<0.001
eGFR	1052	88.95 (70–99.50)	638	83.60 (63.28–95.45)	194	68.05 (41–88.15)	<0.001	<0.001	<0.001
Sodium, mmol/L	1073	141 (138–143)	650	140 (137–143)	196	139 (136–142)	0.007	<0.001	0.012
Potassium, mmol/L	1073	4.20 (3.90–4.53)	650	4.10 (3.80–4.44)	196	4.10 (3.77–4.50)	<0.001	0.013	0.882
ALT, U/L	1059	30.11 (19–49)	633	35 (22–55.66)	190	38.50 (21–57)	<0.001	0.002	0.780
AST, U/L	1050	33.12 (24–50)	629	39 (29–60.15)	188	41 (28–76.75)	<0.001	<0.001	0.531
AST to ALT ratio	1050	1.13 (0.86–1.58)	628	1.13 (0.88–1.58)	188	1.25 (0.85–1.73)	0.806	0.123	0.178
LDH, U/L	993	282 (220–359)	592	318 (263–420.81)	165	362 (253.50–534)	<0.001	<0.001	0.024
CRP, mg/L	1077	48.65 (17.1–94.9)	650	68.21 (31.34–129.73)	196	100.05 (42.53–181.85)	<0.001	<0.001	0.001
Ferritin, µg/L	1007	409.33 (204.63–802.70)	612	568.50 (296.25–1223.25)	170	659.98 (294.24–1402.31)	<0.001	<0.001	0.288
IL-6, ng/L	958	26.15 (13.57–48.5)	581	32.40 (15.80–61.24)	161	36.90 (14.5–77.1)	<0.001	0.003	0.454
D-dimer, µg/L	990	420 (260– 726.25)	614	522.50 (320–926.25)	179	780 (365–1745)	<0.001	<0.001	<0.001
Troponin I, ng/L	899	7.97 (4–15)	561	10 (6–22)	161	20.20 (8.25–116)	<0.001	<0.001	<0.001

ALT—alanine aminotransferase; AST—aspartate aminotransferase; CRP—C-reactive protein; IL-6—interleukin 6; IQR—interquartile range; LDH—lactate dehydrogenase; NLR—neutrophil-to-lymphocyte ratio; WBC—white blood cell count. 1—*p* value comparing patients with normoglycaemia vs. patients with mild hyperglycaemia. 2—*p* value comparing patients with normoglycaemia vs. patients with intermittent hyperglycaemia. 3—*p* value comparing patients with mild hyperglycaemia vs. patients with intermittent hyperglycaemia.

**Table 4 biomedicines-12-00055-t004:** Hazard ratios for 30-day in-hospital mortality.

Characteristic	In-Hospital Mortality	
	HR (95% CI)	*p*
Normoglycaemia	Reference	
Mild hyperglycaemia	1.62 (1.10–2.39)	0.015
Intermittent hyperglycaemia	3.04 (2.01–4.60)	<0.001
Age in years	1.06 (1.05–1.08)	<0.001
Male gender	1.07 (0.77–1.49)	0.689
Hypertension	0.94 (0.67–1.33)	0.729
Coronary artery disease	1.23 (0.69–2.18)	0.487
Congestive heart failure	1.97 (1.33–2.93)	0.001
Obesity	2.94 (1.56–5.57)	0.001
Previous stroke	3.86 (1.95–7.65)	<0.001
Chronic kidney disease	0.61 (0.36–1.04)	0.069
Antivirals (Remdesivir)	0.56 (0.37–0.85)	0.006
Systemic steroids	1.36 (0.94–1.99)	0.108

## Data Availability

From Corresponding Author.

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
