# Peer review of "Hyperglycaemia and Its Prognostic Value in Patients with COVID-19 Admitted to the Hospital in Lithuania"

_biomedicines, 2023, doi:10.3390/biomedicines12010055_

Round 1

Reviewer 1 Report

Comments and Suggestions for Authors

This is a highly interesting topic, particularly in the contesxt of the ongoing COVID-19 pandemic. I would recommend considering a comprehensive review of manuscripts that focus on how various countries are addressing the clallenges highlighted in this study, Examinning diverse approaches and outcomes across different regions could provide valuable insights into the global implications of hyperglycemia and related factors. Please review the references, paying special attention to reference number 33, as it appears to be repeated in several sections of the discussion. 

Comments on the Quality of English Language

The manuscript's structure is comprehensible. I only suggest improving the wording of the results and being clearer in the discussion about the final impact of the findings, emphasizing the study's limitations.

Author Response

REVIEWER 1

This is a highly interesting topic, particularly in the context of the ongoing COVID-19 pandemic. I would recommend considering a comprehensive review of manuscripts that focus on how various countries are addressing the challenges highlighted in this study, examining diverse approaches and outcomes across different regions could provide valuable insights into the global implications of hyperglycemia and related factors.

  1. Please review the references, paying special attention to reference number 33, as it appears to be repeated in several sections of the discussion.

Response: Following a thorough review, we have rectified the repetition issue, ensuring that reference 33 is appropriately cited.

  1. Comments on the Quality of English Language. The manuscript's structure is comprehensible. I only suggest improving the wording of the results and being clearer in the discussion about the final impact of the findings, emphasizing the study's limitations.

Response: Thank you for your valuable feedback. We have revised the wording of the results section to enhance clarity. Additionally, we've made an effort to provide a more comprehensive discussion of the final impact of our findings, emphasizing the study's limitations to ensure transparency.

Reviewer 2 Report

Comments and Suggestions for Authors

The authors focused on the Hyperglycemia and its prognostic value in patients with COVID-19 admitted to the hospital in Lithuania. The paper has a very poor presentation. The entire research has 9! page in MDPI format which occupies 2/3 of a page. A lot of work must be added to give to this manuscript a publishable shape and content. Please see my main suggestions below:

The Authors must check the Instructions for Authors and APPLY them. Nothing is respected.

L1. Type of the paper must be mentioned.

Affiliations and emails must be written in other way – check the Instructions of authors or any of the published paper. 1st page of the blank draft MUST be respected.

Introduction is much too short.

Aim of the study is too poor. It must be addressed from the perspective of describing the contribution to the field under analysis and the elements of scientific novelty presented, as the LAST, SEPARATE paragraph of this section, to be easier visible. Develop it better. What differentiate your paper from other in the same topic? Give a reason for interest in this paper.

L82. How many patients?

2.1. subsection. A detailed CONSORT flow chart must be provided (Figure 1 must be moved here and reedited in a clearest shape).

2.3. A section represents an idea very well developed, not 2 words (not even a sentence)!

Subsection 2.4. All computer programs/softs used, and their variants must be referenced.

Check the entire manuscript. l, ml, etc. should be corrected as L, mL, etc., Litter being the international unit of measure for volume. Be consistent in denotation!

Discuss also the therapeutic measures taken in these patients and the impact of these treatments on hyperglycaemia. How many patients received corticosteroids? Did they develop hyperglycaemia? What about antivirals? Did patients with hyperglycaemia responded less to antivirals? I suggest  checking and referring to https://doi.org/10.1016/j.scitotenv.2021.152072 https://doi.org/10.1016/j.biopha.2022.113089  doi: 10.1016/j.biopha.2022.112700. https://doi.org/10.3892/etm.2021.10080  The analysis is somehow reasonable, however more improvement can be brought by analysis of the treatment of these patients and then making a multiple linear regression that includes variables such as corticosteroid use, antivirals, to see whether hyperglycaemia still represents a prognostic factor.

Conclusions section must be improved.

References must provide all info requested by the Instructions for authors – write them in the MDPI style.

Comments on the Quality of English Language

Moderate English revision

Author Response

REVIEWER 2

The authors focused on the Hyperglycemia and its prognostic value in patients with COVID-19 admitted to the hospital in Lithuania. The paper has a very poor presentation. The entire research has 9! page in MDPI format which occupies 2/3 of a page. A lot of work must be added to give to this manuscript a publishable shape and content.

Please see my main suggestions below:

  1. The Authors must check the Instructions for Authors and APPLY them. Nothing is respected.

Response: Thank you for bringing this to our attention. We have thoroughly reviewed the Instructions and are committed to ensure strict compliance.

  1. Type of the paper must be mentioned.

Response: Thank you for your comment. The type of the paper was specified as an article.

  1. Affiliations and emails must be written in other way – check the Instructions of authors or any of the published paper. 1st page of the blank draft MUST be respected.

Response: We have reviewed the Instructions for Authors and revised affiliations and emails according to the specified format. The first page of the blank draft has been appropriately formatted.

  1. Introduction is much too short.

Response: Thank you for your comment. The introduction has been expanded to provide a more comprehensive overview.

  1. Aim of the study is too poor. It must be addressed from the perspective of describing the contribution to the field under analysis and the elements of scientific novelty presented, as the LAST, SEPARATE paragraph of this section, to be easier visible. Develop it better. What differentiate your paper from other in the same topic? Give a reason for interest in this paper.

Response: We appreciate your valuable advice. The study aim has been expanded to include a last, separate paragraph that explicitly highlights the contribution to the field, scientific novelty, and what sets our paper apart from others on the same topic.

  1. How many patients?

Response: More details, including the number of patients, have been added to the introduction.

  1. 1. subsection. A detailed CONSORT flow chart must be provided (Figure 1 must be moved here and reedited in a clearest shape).

Response: Thank you for your comment. Figure 1 has been reedited into a CONSORT flow chart and moved to section 2.1.

  1. 3. A section represents an idea very well developed, not 2 words (not even a sentence)!

Response: The section has been amended with a more detailed outcome description.

  1. Subsection 2.4. All computer programs/softs used, and their variants must be referenced.

Response: We have added clarification that the analysis was performed using IBM Statistical Package for the Social Sciences software version 20.0, and Microsoft Excel was used to produce figures.

  1. Check the entire manuscript. l, ml, etc. should be corrected as L, mL, etc., Litter being the international unit of measure for volume. Be consistent in denotation!

Response: The entire manuscript has been checked, and units have been updated in Tables 1 and 3, Figures 1, 2, and 3, and the manuscript text to ensure consistency.

  1. Discuss also the therapeutic measures taken in these patients and the impact of these treatments on hyperglycaemia. How many patients received corticosteroids? Did they develop hyperglycaemia? What about antivirals? Did patients with hyperglycaemia responded less to antivirals? I suggest checking and referring to https://doi.org/10.1016/j.scitotenv.2021.152072 https://doi.org/10.1016/j.biopha.2022.113089  doi: 10.1016/j.biopha.2022.112700. https://doi.org/10.3892/etm.2021.10080 

Response. Results section was updated to include results related to systemic steroids use and treatment with Remdesivir. No other antiviral medications were used to treat COVID-19. The discussion was amended, and reference to the suggested articles was added.

  1. The analysis is somehow reasonable, however more improvement can be brought by analysis of the treatment of these patients and then making a multiple linear regression that includes variables such as corticosteroid use, antivirals, to see whether hyperglycaemia still represents a prognostic factor.

Response. Thank you for your valuable suggestion. We conducted a more detailed analysis by performing a multivariable binary logistic regression model to explore the association between glucose levels and the need for invasive mechanical ventilation (IMV). In this model, the need for IMV served as the dependent variable, with age, gender, comorbidities showing statistically significant differences in proportions between patients’ groups, the use of systemic steroids, and treatment with Remdesivir as predictors. The regression analysis revealed that the use of systemic steroids and treatment with Remdesivir did not show a significant association with the need for IMV. The odds ratios for IMV in patients with intermittent hyperglycemia and mild hyperglycemia remained higher compared to patients presenting with normal glucose levels at admission. Furthermore, the identified other risk factors associated with the need for IMV, namely male gender, obesity, and congestive heart failure, remained statistically significant. Results of the multivariable binary logistic regression are provided in the table below. Since our focus was on the association of risk factors present at admission with the need for IMV, we did not include this analysis in the article.

Table. Risk factors and Odds Ratios (OR) associated with the need of invasive mechanical ventilation in COVID-19 patients without diabetes.

Characteristic

Invasive mechanical ventilation

OR (95% CI)

p

Normoglycaemia

Reference

Mild hyperglycaemia

1.98 (1.19 – 3.29)

0.009

Intermittent hyperglycaemia

4.73 (2.64 – 8.47)

<0.001

Age in years

1.01 (0.99 – 1.03)

0.220

Male gender

1.63 (1.04 – 2.56)

0.034

Hypertension

1.23 (0.77-1.96)

0.393

Coronary artery disease

1.66 (0.63 – 4.39)

0.309

Congestive heart failure

2.22 (1.13 – 4.35)

0.020

Obesity

5.77 (2.82 – 11.80)

<0.001

Previous stroke

0

0.998

Chronic kidney disease

0.82 (0.39 – 2.12)

0.904

Antivirals (Remdesivir)

0.73 (0.44 – 1.22)

0.233

Systemic steroids

1.33 (0.80 – 2.21)

0.208

  1. Conclusions section must be improved.

Response: Thank you for the valuable feedback. We have carefully reviewed and enhanced the conclusions section.

  1. References must provide all info requested by the Instructions for authors – write them in the MDPI style.

Response: We have meticulously reviewed the references and have taken steps to correct any deviations from the Instructions for authors.

Reviewer 3 Report

Comments and Suggestions for Authors

This papeer is a comprehensive study that investigates the association between hyperglycemia at admission and the outcomes of COVID-19 in patients without pre-existing diabetes. The study is retrospective and observational.

Major issues:

Methodological Detailing: The methods section, while adequate, could benefit from more detailed explanations. Specifics regarding patient selection criteria, data collection procedures, and statistical analyses should be elaborated to enhance reproducibility and understanding.

Limitations Addressing: The paper could provide a more comprehensive discussion of its limitations. Addressing potential biases in more depth, especially concerning the retrospective nature of the study and the generalizability of the findings, would strengthen the paper.

Comparative Analysis: The study primarily focuses on hyperglycemia in non-diabetic COVID-19 patients. A comparative analysis with diabetic patients might have provided additional insights into the unique impacts of hyperglycemia in these different cohorts. Results should also be evaluated in relation to the effects of reduced physical activity on the lipid profile in patients at high cardiovascular risk during covid-19 

Long-term Outcomes: The paper focuses on in-hospital outcomes. Including or discussing the potential long-term effects of hyperglycemia in recovered COVID-19 patients could provide a more holistic view of the disease's trajectory.

Broader Contextualization: While the study is well-rooted in its clinical context, integrating it more explicitly with existing literature on COVID-19 and metabolic disturbances could enhance its contribution to the broader field. This would involve a more detailed comparison with other studies and a discussion on how these findings fit into the current understanding of COVID-19.

Critical issue:

there is a lot of smilitude with this paper, check for plagiarism (seee also attachment):

Kubiliute I, Vitkauskaite M, Urboniene J, Svetikas L, Zablockiene B, Jancoriene L (2023) Clinical characteristics and predictors for in- hospital mortality in adult COVID-19 patients: A retrospective single center cohort study in Vilnius, Lithuania. PLoS ONE 18(8): e0290656.
https://doi. org/10.1371/journal.pone.0290656

Comments on the Quality of English Language

The paper is generally well-written with clear language. However, minor edits for grammar, punctuation, and consistency in style could enhance its readability and overall presentation.

Author Response

REVIEWER 3

This paper is a comprehensive study that investigates the association between hyperglycemia at admission and the outcomes of COVID-19 in patients without pre-existing diabetes. The study is retrospective and observational.

Major issues:

  1. Methodological Detailing: The methods section, while adequate, could benefit from more detailed explanations. Specifics regarding patient selection criteria, data collection procedures, and statistical analyses should be elaborated to enhance reproducibility and understanding.

Response: Thank you for valuable comment. We have diligently reviewed the specified sections and incorporated additional details on selection of patients, data collection procedures, and the programs used for analysis.

  1. Limitations Addressing: The paper could provide a more comprehensive discussion of its limitations. Addressing potential biases in more depth, especially concerning the retrospective nature of the study and the generalizability of the findings, would strengthen the paper.

Response: We appreciate the reviewer's insightful comments and acknowledge the need for a more comprehensive discussion of the study's limitations. In response, we expanded the limitations section to delve more deeply into the inherent biases associated with retrospective study limited to single centre.

  1. Comparative Analysis: The study primarily focuses on hyperglycemia in non-diabetic COVID-19 patients. A comparative analysis with diabetic patients might have provided additional insights into the unique impacts of hyperglycemia in these different cohorts. Results should also be evaluated in relation to the effects of reduced physical activity on the lipid profile in patients at high cardiovascular risk during covid-19.

Response: We appreciate the thoughtful feedback from the reviewer. While we acknowledge the potential insights gained from a comparative analysis between diabetic and non-diabetic COVID-19 patients, the primary focus of our study was to investigate hyperglycemia in the non-diabetic population. Regarding the effects of reduced physical activity on the lipid profile in patients at high cardiovascular risk during COVID-19, we regret to inform you that our study lacks data on physical activity, and a lipid profile is available for only a very few patients, which does not allow for a proper analysis. We appreciate these considerations and are committed to exploring potential ways to incorporate them into our future research.

  1. Long-term Outcomes: The paper focuses on in-hospital outcomes. Including or discussing the potential long-term effects of hyperglycemia in recovered COVID-19 patients could provide a more holistic view of the disease's trajectory.

Response: We appreciate the suggestion. We have updated the discussion section to incorporate insights into the potential long-term implications of hyperglycemia beyond the in-hospital outcomes. We hope that this addition enhances the overall depth and understanding of our study.

  1. Broader Contextualization: While the study is well-rooted in its clinical context, integrating it more explicitly with existing literature on COVID-19 and metabolic disturbances could enhance its contribution to the broader field. This would involve a more detailed comparison with other studies and a discussion on how these findings fit into the current understanding of COVID-19.

Response: Thank you for the valuable suggestion to integrate our study more explicitly with existing literature on COVID-19 and metabolic disturbances. We have made efforts to enhance the broader contextualization of our study by providing a more detailed comparison with other relevant studies and discussing how our findings contribute to the current understanding of COVID-19.

  1. Critical issue: There is a lot of similitude with this paper, check for plagiarism (see also attachment): Kubiliute I, Vitkauskaite M, Urboniene J, Svetikas L, Zablockiene B, Jancoriene L (2023) Clinical characteristics and predictors for in- hospital mortality in adult COVID-19 patients: A retrospective single center cohort study in Vilnius, Lithuania. PLoS ONE 18(8): e0290656. https://doi. org/10.1371/journal.pone.0290656. peer-review-33620894.v1.pdf

Response: We acknowledge the similarities with the referenced paper and want to clarify that both studies stem from the same database and research team. While they share a common database, each paper explores a different angle and aspect of the data, contributing distinct findings to the scientific discourse. We have duly added the referenced paper to our citations to highlight this relationship.

  1. Comments on the Quality of English Language

The paper is generally well-written with clear language. However, minor edits for grammar, punctuation, and consistency in style could enhance its readability and overall presentation.

Response: Thank you for your positive feedback on the overall clarity of the paper. In response, we have thoroughly reviewed and implemented the necessary corrections to enhance the readability and overall presentation of the manuscript.

Round 2

Reviewer 2 Report

Comments and Suggestions for Authors

Accept in current form.

Comments on the Quality of English Language

Accept in current form.

Reviewer 3 Report

Comments and Suggestions for Authors

The authors set up the paper well following my instructions

Comments on the Quality of English Language

English is fine